# Genetic Association between the Levels of Plasma Lipids and the Risk of Aortic Aneurysm and Aortic Dissection: A Two-Sample Mendelian Randomization Study

**DOI:** 10.3390/jcm12051991

**Published:** 2023-03-02

**Authors:** Rui Li, Chao Zhang, Xinling Du, Shi Chen

**Affiliations:** 1Department of Cardiovascular Surgery, Union Hospital, Tongji Medical College, Huazhong University of Science and Technology, Wuhan 430022, China; 2State Key Laboratory of Cardiovascular Disease, Fuwai Hospital, National Center for Cardiovascular Diseases, Chinese Academy of Medical Sciences and Peking Union Medical College, Beijing 100730, China

**Keywords:** plasma lipids, aortic aneurysm, aortic dissection, Mendelian randomization analysis

## Abstract

Although a growing number of studies have attempted to uncover the relationship between plasma lipids and the risk of aortic aneurysm (AA), it remains controversial. Meanwhile, the relationship between plasma lipids and the risk of aortic dissection (AD) has not been reported on. We conducted a two-sample Mendelian randomization (MR) analysis to evaluate the potential relationship between genetically predicted plasma levels of lipids and the risk of AA and AD. Summary data on the relationship between genetic variants and plasma lipids were obtained from the UK Biobank and Global Lipids Genetics Consortium studies, and data on the association between genetic variants and AA or AD were taken from the FinnGen consortium study. Inverse-variance weighted (IVW) and four other MR analysis methods were used to evaluate effect estimates. Results showed that genetically predicted plasma levels of low-density lipoprotein cholesterol, total cholesterol, or triglycerides were positively correlated with the risk of AA, and plasma levels of high-density lipoprotein cholesterol were negatively correlated with the risk of AA. However, no causal relationship was found between elevated lipid levels and the risk of AD. Our study revealed a causal relationship between plasma lipids and the risk of AA, while plasma lipids had no effect on the risk of AD.

## 1. Introduction

Aortic aneurysm (AA) is the second most common aortic disease after atherosclerosis with a high risk of sudden death characterized by localized progressive and irreversible full-thickness dilation of the aorta [1]. The incidence of abdominal aortic aneurysm (AAA) increases significantly in men older than 55 years of age and in women older than 70 years of age, and it is four to five times as common in men as in women. As for thoracic aortic aneurysm (TAA), its overall rate of incidence is about 5 to 10 cases per 100,000 person-years, and the incidence increases as the population ages [2]. The morbidity of TAA is higher in men than in women, but women have worse outcomes [3]. The risk of death from aneurysm rupture is extremely high. The mortality rate of AAA patients is 60–70%, and about 150,000 to 200,000 people die from AAA rupture each year [4]. Aortic dissection (AD) is a cardiovascular emergency with high mortality and poor prognosis, which is characterized by tearing of the aortic intima and blood entering the vascular media. The overall incidence of acute AA is 7.7 cases per 100,000 person-years [5]. About two-thirds of patients with aortic dissection are male [6]. In view of the high mortality risk of AA and AD, it is very important to explore their risk factors and carry out primary prevention to reduce the incidence and harm of AA and AD.

Dyslipidemia refers to an imbalance in plasma levels of cholesterol and triglyceride, including total cholesterol (TC), low-density lipoprotein cholesterol (LDL-C), high-density lipoprotein cholesterol (HDL-C), and triglyceride (TG), and manifests as elevated plasma concentrations of TC, LDL-C, or TG, or a low plasma concentration of HDL-C or a combination of these features [7]. Numerous studies have shown that dyslipidemia is a risk factor for a variety of cardiovascular diseases [8]. A growing number of observational studies and prospective cohort studies have attempted to uncover the relationship between plasma lipids and the risk of AA [9,10,11,12]. However, whether there is a causal relationship between the levels of various lipid types and the risk of AA remains controversial. Observational studies found obvious dyslipidemia in AD patients, and the in-hospital mortality of AD patients was correlated with dyslipidemia [13,14]. However, whether dyslipidemia is a risk factor for the occurrence of AD has not been reported on.

In recent years, due to the susceptibility of traditional observational studies to residual confounding effects and reverse causality, Mendelian randomization (MR) analysis has been widely used to evaluate the potential causality between various exposures and clinical outcomes by using genetic variants as instrumental variables (IVs) for exposures. Because the genetic variation occurs during conception and precedes the onset of disease, MR analysis can overcome reverse causation bias, reduce potential unmeasured confounders, and improve causal inference [15]. The randomized controlled trial (RCT) is considered the gold standard for causal inference. However, sometimes RCTs cannot be carried out due to reasons such as ethical issues, impracticality, and high expenditure of research funds. When RCTs are not possible, evidence from MR analysis is the highest level of evidence for causal inference [16]. Several MR studies have found that plasma lipids are causally associated with the risk of AA, but there is still much controversy surrounding their findings [17,18,19]. Due to the low incidence and high mortality risk of AD, it is difficult to investigate the causal relationship between plasma lipids and AD through large-scale prospective studies. Therefore, an MR study is an ideal and feasible alternative to study the causal relationship between plasma lipids and AD.

In this study, we aimed to apply a two-sample MR analysis to investigate the role of genetically predicted plasma LDL-C, TC, TG, and HDL-C levels in the risk of AA and AD.

## 2. Materials and Methods

### 2.1. Study Design

We conducted a two-sample Mendelian randomization (MR) analysis to clarify the potential causal effects of genetically predicted plasma LDL-C, TC, TG, and HDL-C levels on AA and AD (Figure 1). Three key assumptions of our MR design are as follows: (1) single nucleotide polymorphisms (SNPs) are robustly associated with plasma LDL-C, TC, TG, or HDL-C; (2) SNPs are independent of other known confounders; (3) SNPs affect the occurrence of AA or AD only through plasma LDL-C, TC, TG, or HDL-C.

### 2.2. Data Sources

The analysis was carried out using published summary statistics from publicly available genome-wide association studies (GWASs), mainly on European individual characteristics, both male and female (Table 1). The UK Biobank is a prospective study that enrolled more than half a million participants aged 40 to 69 years between 2006 and 2010. Broad phenotypic and genotypic details of the participants have been collected, including genome-wide genotyping and data on a broad range of health-related outcomes [20]. The Global Lipids Genetics Consortium (GLGC) study is a genome-wide genotyping analysis of 188,577 individuals to search for loci associated with plasma lipid levels [21]. The aim of the FinnGen study is to research the national health registry data and genome of 500,000 Finns. Currently, it has reached 224,737 phenotyped and genotyped individuals, reported a genome-wide association study of 1932 clinical endpoints derived from national health registries, and found genome-wide significant associations at 2491 independent loci. GWASs summary statistics for plasma levels of LDL-C (*n* = 440,546), TG (*n* = 441,016), and HDL-C (*n* = 403,943) were obtained from the UK Biobank study, and plasma levels of TC (*n* = 187,365) were obtained from the GLGC study, which assessed the relationship between 4 lipid traits and the SNPs. SNPs associated with AA (*n* = 209,366) or AD (*n* = 207,011) were analyzed from FinnGen consortium data. Because the GWASs data are all publicly available and have been approved by the appropriate ethical review boards, no additional ethical approval was required for the analysis in this study.

### 2.3. Selection and Validation of SNPs

Independent SNPs associated with plasma levels of LDL-C, TC, TG, or HDL-C were identified according to three criteria. First, SNPs that had reached a genome-wide significance level (*p* < 5 × 10^−8^) were selected. Second, the independence of these selected SNPs was evaluated based on the linkage disequilibrium. SNPs were deleted if they were in the linkage disequilibrium (r^2^ > 0.01) and in proximity (clumping window of 1 Mb) to other SNPs with higher *p*-values. Third, SNPs with F-statistics greater than ten were selected to mitigate the impact of potential bias.

### 2.4. Two-Sample MR Analysis

Five two-sample MR analysis methods, including inverse-variance weighted (IVW), MR-Egger, weighted median, simple mode, and weighted mode, were applied in our study. Among the five methods, the IVW method was used as the primary analysis method to evaluate the effect estimates. Cochran’s Q test was applied to test the heterogeneity of the IVs, and significant heterogeneity was considered to exist when *p* < 0.05. If heterogeneity was detected among the IVs, random-effects IVW was used; otherwise, fixed-effects IVW was applied. The intercept of MR-Egger regression was used to detect whether the estimate of causality was affected by the pleiotropic effects of the genetic variants. The leave-one-out analysis was performed by removing each instrumental variable in turn to determine whether the result was disproportionately affected by a single SNP.

Two-sided *p* value < 0.05 was considered as statistically significant. All statistical analyses were performed using the “TwoSampleMR” packages in R Software (version 4.2.0).

## 3. Results

### 3.1. SNP Selection and Validation

Independent SNPs included in our study as IVs are shown in Appendix A.

### 3.2. Estimates of Causality between Plasma LDL-C, TC, TG, and HDL-C Levels and the Risk of AA

The IVW analysis showed that elevated genetically predicted plasma levels of LDL-C (odds ratio (OR), 1.47; 95% confidence interval (CI), 1.20–1.81; *p* = 0.006), TC (OR, 1.30; 95% CI, 1.07–1.56; *p* = 0.007), or TG (OR, 1.31; 95% CI, 1.13–1.53; *p* < 0.001) were associated with an increased risk of AA. Meanwhile, an elevated plasma level of HDL-C (OR, 0.71; 95% CI, 0.61–0.82; *p* < 0.001) was associated with a decreased risk of AA (Figure 2). These associations remained consistent across the other four MR analysis methods, indicating the robustness of the main results, although 95% CIs were wide in some methods. The intercept of MR-Egger regression showed that no evidence of pleiotropy between SNPs was observed (Table 2). The scatter plots of the association between plasma lipids and AA are presented in Appendix A. The IVs of HDL-C, LDL-C, TC, and TG were found to have significant heterogeneity using Cochran’s Q test (Table 2), so the IVW method with random effects was applied in our study to mitigate the influence of heterogeneity. The associations are shown in the funnel plots (Appendix A). The leave-one-out analyses showed that the relationship between genetically predicted plasma levels of HDL-C, LDL-C, TC, or TG and the risk of AA was not driven by any individual SNP (Appendix A). Considering the effect of gender on the risk of AA, we subsequently stratified the data by gender and found that gender had no effect on the findings of our study (Appendix A).

### 3.3. Estimates of Causality between Plasma LDL-C, TC, TG and HDL-C Levels and the Risk of AD

The IVW analysis revealed that the genetically predicted plasma level of HDL-C (OR, 0.90; 95% CI, 0.64–1.26; *p* = 0.542), LDL-C (OR, 1.03; 95% CI, 0.69–1.56; *p* = 0.872), TC (OR, 1.14; 95% CI, 0.81–1.60; *p* = 0.461), or TG (OR, 0.77; 95% CI, 0.23–2.55; *p* = 0.502) was not associated with the risk of AD (Figure 3). Given that gender is also a risk factor for AD, we then stratified the data by gender and found that gender had no effect on our findings (Appendix A).

## 4. Discussion

In this two-sample MR analysis, we investigated the genetic association between levels of plasma lipids and the risk of AA and AD. Our study provided new evidence for the previous clinical findings that elevated plasma levels of LDL-C, TC, or TG were associated with an increased risk of AA, and an elevated plasma level of HDL-C was inversely associated with the risk of AA. In addition, we revealed for the first time that, based on the same GWASs databases, there was no causal relationship between genetically predicted lipid levels and the risk of AD.

Current studies on the relationship between plasma lipids and the risk of AA are still controversial. An observational study revealed that LDL-C, TC, TG, and HDL-C were independently associated with the risk of AAA [9]. However, some prospective cohort studies have come to very different conclusions. A prospective cohort study of a large multiethnic population in northern California showed that a high serum TC level was a significant predictor of the risk of AAA [11]. A 7-year prospective study from Norway has suggested that only those with serum TC higher than 7.55 mmol/L had an increased risk of AA, and that the HDL-C level was inversely associated with the risk of developing an AAA [10]. Golledge et al. found that serum TG or LDL-C concentrations had no relationship with the risk of having an AAA in men aged 65 years or older [22]. Because traditional observational studies, even well-designed prospective studies with large samples, are susceptible to residual confounding effects and reverse causality, several studies have tried to use MR analysis to explore the relationship between plasma lipids and the risk of AA. In MR analysis, plasma lipid levels are genetically determined and therefore free from the influence of lifestyle and other confounding factors. However, depending on the genetic information of the population in a given database, selection of a different database may lead to different analysis results. Based on the GLGC study and five international AAA GWASs databases, Harrison et al. concluded that HDL-C, LDL-C, and TG were all associated with AAA, but the relationship between TC and the risk of AAA was not investigated in their study [17]. Analyses using the GLGC study and data on European Americans (EAs) from the Atherosclerosis Risk in Communities (ARIC) study showed that HDL-C and TG were not associated with AAA. TC was confirmed to be associated with the risk of AAA, but the association between LDL-C and AAA risk was less consistent because of the wider confidence intervals [19]. Consistent with our findings, a study from Chen et al. found that genetically predicted plasma levels of LDL-C, TC, or TG were positively correlated with AA, and the level of HDL-C was negatively correlated with the risk of AA, based on the GLGC and UK Biobank studies [18]. Using the same databases, Allara et al. reported a positive association between the LDL-C level and the risk of AAA, and an inverse association between the HDL-C level and the risk of AAA. TG was not associated with AAA, and HDL-C, HDL-C, and TG were not associated with TAA [23]. Studies have indicated that atherosclerosis is the main risk factor for AAA, while connective tissue disease and bicuspid aortic valve are the most common risk factors for TAA [2]. The FinnGen consortium database used in our study did not distinguish between AAA and TAA patients, so different types of AA could not be evaluated separately. In all of the above studies, lipid-associated SNPs were identified based on the GLGC study. In our study, GWASs summary statistics for LDL-C, TG, and HDL-C were obtained from the UK Biobank study, which has a sample size more than twice that of the GLGC. Based on the UK Biobank, GLGC, and FinnGen databases, our study provided new evidence for the nature of the causal relationship between plasma lipids and the risk of AA. Overall, our study and previous studies have shown that the levels of plasma lipids are closely related to the risk of AA, which provides a theoretical basis for the prevention of AA by controlling the levels of plasma lipids.

At present, there are no studies on the association between plasma lipids and the risk of AD. Using MR analysis, we found, for the first time, that genetically determined plasma levels of HDL-C, LDL-C, TC, and TG were not associated with the risk of AD. Using the same IVs from the UK Biobank and GLGC studies and the same population from the FinnGen consortium database, there were positive results for AA but negative results for AD, which greatly enhanced the reliability of our findings on estimates of causality between genetically determined plasma lipid levels and the risk of AD and indicated that the pathogenesis of AA and AD were very different. An observational study found that patients with AD combined with basic atherosclerotic diseases had significantly higher levels of LDL-C, TC, and TG and significantly lower levels of HDL-C in their plasma than patients with basic atherosclerotic diseases alone [13]. There are limitations to observational studies, however, so this study cannot show a cause-and-effect relationship between dyslipidemia and the risk of AD. Due to the low incidence (2.6 to 3.5 cases per 100,000 persons per year) and sudden onset of AD, no prospective cohort studies on the levels of plasma lipids and the risk of AA have been conducted [24]. Meanwhile, it is also difficult to investigate the relationship between plasma lipids and AD using an experimental study because there is no suitable animal model of AD at present. Previous studies on the relationship between dyslipidemia and AD mainly focused on the association between dyslipidemia and in-hospital mortality in patients with acute AD (AAD). Liu et al. found that a low plasma TC level (<4.00 mmol/L) was associated with an increased in-hospital mortality in patients with type-A AAD, which may be due to the poor nutritional status of patients with low TC leading to poor tolerance to AD [25]. A study from Lin et al. reported that serum TG/HDL-C levels are positively correlated with in-hospital mortality in patients with type-A AAD [14]. As for patients with type-B AAD, no significant association was found between TC and in-hospital mortality, and the TG/HDL-C ratio is positively correlated with in-hospital mortality only when the ratio is less than 2.05 [25,26]. In conclusion, although there is no causal relationship between plasma lipids and the risk of AD, dyslipidemia may increase in-hospital mortality in patients with AD. Therefore, attention should still be paid to the screening, diagnosis, and management of dyslipidemia.

A previous two-sample Mendelian randomization study, based on the GLGC and UK Biobank databases, assessed whether there was a gender-dependent difference in the effect of genetically determined plasma levels of LDL-C on the risk of AA and AD [27]. In the study, AA and AD were combined as one outcome and not analyzed separately, and the results showed no gender-specific effect in the association between genetically determined plasma levels of LDL-C and AA and AD risk, which is similar to the results of our study.

Lipid-lowering therapy is considered to play an important role in the prevention of a variety of cardiovascular diseases, such as coronary heart disease, myocardial infarction, and stroke [28,29,30,31]. However, there are only a few studies on whether lipid-lowering therapy can prevent AA. A meta-analysis and systematic review showed that statin therapy could reduce the risks of AAA rupture, post-repair mortality, all-cause mortality, and adverse events, but not the risk of AAA incidence [32]. Proprotein convertase subtilisin/Kexin type 9 (PCSK9) can regulate cholesterol metabolism by degrading the low-density lipoprotein receptor [33]. An observational study found that a loss-of-function mutation in PCSK9 has a protective effect on AAA [34]. Meanwhile, animal experiments showed that a PCSK9 gain-of-function mutation promoted AAA occurrence in mice [35]. PCSK9 inhibitors, Evolocumab and Alirocumab, have been approved for the clinical treatment of hyperlipidemia [36,37]. However, whether Evolocumab or Alirocumab have a protective effect on AAA remains to be studied. In conclusion, further studies are currently needed to determine whether lipid-lowering therapy can prevent the occurrence of AA. As for AD, no clinical or experimental studies have reported the relationship between lipid-lowering therapy and the risk of AD.

In our study, we used a two-sample MR analysis based on the large-scale GWASs databases to evaluate the causal effect of genetically predicted plasma lipid levels on the risk of AA and AD. Compared with traditional observational studies, our study overcame the influence of residual confounding effects and reverse causality, which enhanced the reliability of our findings. Using different databases from those used in previous studies, our study provided new evidence for the nature of the causal relationship between genetically determined lipid levels and the risk of AA. Meanwhile, we revealed for the first time that genetically determined lipid levels are not associated with AD risk. However, several limitations of our study should be considered. First, all large-scale GWASs data used in our study were obtained from the European population. Therefore, whether our findings are applicable to other ethnic groups requires further study. Second, the data on association between the genetic variant and AD or AA were taken from the FinnGen consortium database. Aortic aneurysms and dissections were not classified into different types in this database, so the effect of dyslipidemia on different types of aneurysms and dissections could not be obtained. Finally, although no evidence of pleiotropic effects was found in the MR-Egger intercept tests from our study, the potential influence of directional pleiotropy is difficult to completely exclude.

In conclusion, our study revealed a causal relationship between genetically determined plasma lipid levels and the risk of AA, and found for the first time that genetically determined plasma lipid levels had no effect on the risk of AD, which may help to guide clinical prevention, clinical trial design, and risk-factors analysis of AA and AD.

## Figures and Tables

**Figure 1 jcm-12-01991-f001:**
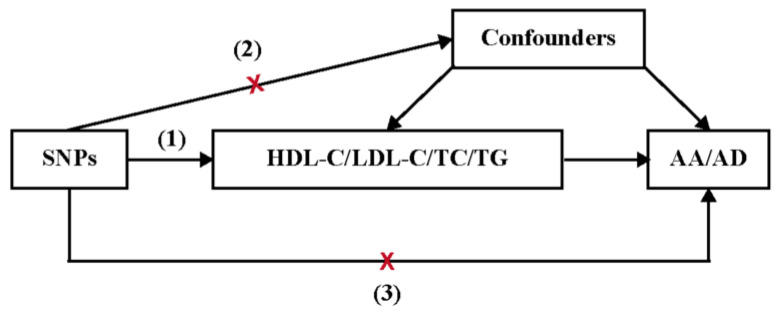
Three key assumptions of the MR study. (1) SNPs are robustly associated with plasma LDL-C, TC, TG, or HDL-C; (2) SNPs are independent of other known confounders; (3) SNPs affect the risk of AA or AD only through plasma LDL-C, TC, TG, or HDL-C. The red X means that the SNPs selected as the instrumental variables are not associated with the confounders and the outcomes directly.

**Figure 2 jcm-12-01991-f002:**
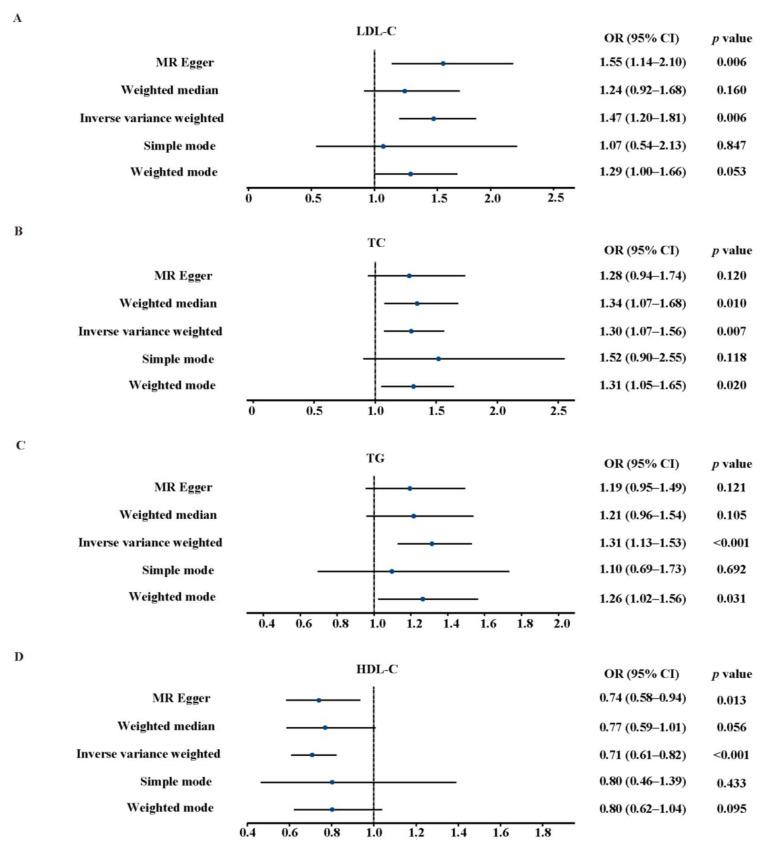
Associations between genetically predicted plasma levels of LDL-C (**A**), TC (**B**), TG (**C**) and HDL-C (**D**) and the risk of AA. LDL-C, low-density lipoprotein cholesterol; TC, total cholesterol; TG, triglyceride; HDL-C, high-density lipoprotein cholesterol; CI, confidence interval; OR, odds ratio.

**Figure 3 jcm-12-01991-f003:**
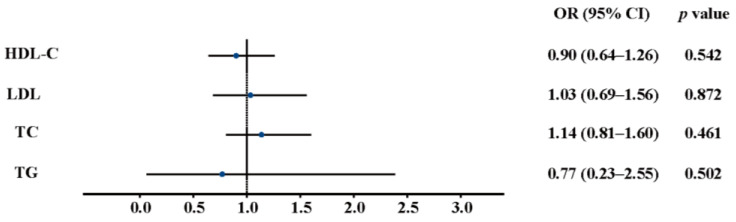
Associations between the genetically predicted plasma level of LDL-C, TC, TG and HDL-C and the risk of AD. LDL-C, low-density lipoprotein cholesterol; TC, total cholesterol; TG, triglyceride; HDL-C, high-density lipoprotein cholesterol; CI, confidence interval; OR, odds ratio.

**Table 1 jcm-12-01991-t001:** Baseline characteristics of lipids, aortic aneurysm, and aortic dissection datasets.

Trait	Year	Consortium	Population	Sample Size	*n* Case	*n* Control	*n* SNPs
HDL-C	2020	UK Biobank	European	403,943			1,232,1875
LDL-C	2020	UK Biobank	European	440,546			1,232,1875
TC	2013	GLGC	European (~96%), non-European (~4%)	187,365			2,446,982
TG	2020	UK Biobank	European	441,016			12,321,875
AA	2021	FinnGen	European	209,366	2825	206,541	16,380,417
AD	2021	FinnGen	European	207,011	470	206,541	16,380,411

HDL-C, high-density lipoprotein cholesterol; LDL-C, low-density lipoprotein cholesterol; TC, total cholesterol; TG, triglycerides; AA, aortic aneurysm; AD, aortic dissection; SNP, single nucleotide polymorphism; GLGC, Global Lipids Genetics Consortium.

**Table 2 jcm-12-01991-t002:** Tests of pleiotropy of selected SNPs and heterogeneity between SNPs.

Risk Factors	Pleiotropy Test		Heterogeneity Test	
	Beta (SE)	*p* Value	Cochran’s Q	*p* Value
HDL-C	0.003	0.636	383.72	0.002
LDL-C	0.006	0.668	238.12	<0.001
TC	0.009	0.917	159.17	<0.001
TG	0.004	0.254	347.26	<0.001

HDL-C, high-density lipoprotein cholesterol; LDL-C, low-density lipoprotein cholesterol; TC, total cholesterol; TG, triglycerides.

## Data Availability

The data presented in our study are included in this article, and further inquiries can be directed to S.C., fwh_cs@163.com or xinlingdu@hust.edu.cn.

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
