# Peer review of "Genetic Association between the Levels of Plasma Lipids and the Risk of Aortic Aneurysm and Aortic Dissection: A Two-Sample Mendelian Randomization Study"

_jcm, 2023, doi:10.3390/jcm12051991_

Round 1
Reviewer 1 Report
The manuscript by Rui Li et al. entitled "Genetic Association between the Levels of Plasma Lipids and 2 the Risk of Aortic Aneurysm and Aortic Dissection: A Two-3 Sample Mendelian Randomization Study" is concise and very well written. They have performed a well-designed two-sample Mendelian randomization analysis to assess the possible relationship between plasma lipids and the risk of AA and AD, showing a causal relationship between plasma lipids and the risk of AA, but not the risk of AD. I have only minor issues regarding the manuscript. I missed some information on the well-known risk factor for AA which is gender (male) and its association with the SNPs evaluated, please include, if possible, these data, if not at least mentioned it in the discussion.
Author Response
We gratefully appreciate for your valuable suggestions and questions. We have revised our manuscript as you suggested. Please see the attachment.

Reviewer 2 Report
In this paper, Rui Li conducted a two-sample Mendelian randomization analysis to evaluate the potential relationship between plasma lipids and the risk of AA and AD, coming to detect a causal relationship between plasma lipids and the risk of AA, while plasma lipids had no effect on the risk of AD.
The study appears to be well conducted and the paper is quite clear in its exposition.
However, some points should be further developed:
- It should be stressed more in the text that plasma lipid levels are the genetically predetermined ones, and not the general ones that could also be influenced by lifestyle and other confounders. Therefore it is important to add "genetically predicted" when talking about plasma lipid levels
- Analysis divided by gender (male/female) would be interesting. Does the observed association hold by gender as well? Also, the number of males and females should be included in Table 1.
- many references are older than 5 years, and should therefore be replaced with some newer ones. Also, many references are repetitive.
- some language errors are present and the writing is often weighted down (such as the beginning of page 2, where "because" is repeated many times)
- page 2, why not synthesize "aortic dissection" with AD? The repetition of the word makes the reading less fluent
- Sentence 167-169 (page 6) with reference [18] should be reworded because it is unclear. In fact, the cited work still showed an association of AAA risk with TC, although the associations were less consistent for LDL cholesterol due to wider confidence intervals.
The work cited is very similar to the one you presented. Therefore, the strengths and major differences that make your study original should be highlighted
- Sentence 224 page 7 is unclear and should be revised
Author Response

(The authors gave the same response as above.)
